# Overexpression of the Bacteriophage T4 *motB* Gene Alters H-NS Dependent Repression of Specific Host DNA

**DOI:** 10.3390/v13010084

**Published:** 2021-01-09

**Authors:** Jennifer Patterson-West, Chin-Hsien Tai, Bokyung Son, Meng-Lun Hsieh, James R. Iben, Deborah M. Hinton

**Affiliations:** 1Gene Expression and Regulation Section, Laboratory of Cell and Molecular Biology, National Institute of Diabetes and Digestive and Kidney Diseases, National Institutes of Health, Bethesda, MD 20892, USA; jennifer.west@nih.gov (J.P.-W.); bokyung.son@nih.gov (B.S.); menglunhsieh@gmail.com (M.-L.H.); 2Center for Cancer Research, Laboratory of Molecular Biology, National Cancer Institute, National Institutes of Health, Bethesda, MD 20892, USA; taic@mail.nih.gov; 3Molecular Genomics Core, Eunice Kennedy Shriver National Institute of Child Health and Human Development, National Institutes of Health, Bethesda, MD 20892, USA; james.iben@nih.gov

**Keywords:** bacteriophage T4, MotB, H-NS, nucleoid, host takeover, DNA-binding protein, RNA-seq, transcriptome analysis

## Abstract

The bacteriophage T4 early gene product MotB binds tightly but nonspecifically to DNA, copurifies with the host Nucleoid Associated Protein (NAP) H-NS in the presence of DNA and improves T4 fitness. However, the T4 transcriptome is not significantly affected by a *motB* knockdown. Here we have investigated the phylogeny of MotB and its predicted domains, how MotB and H-NS together interact with DNA, and how heterologous overexpression of *motB* impacts host gene expression. We find that *motB* is highly conserved among *Tevenvirinae*. Although the MotB sequence has no homology to proteins of known function, predicted structure homology searches suggest that MotB is composed of an N-terminal Kyprides-Onzonis-Woese (KOW) motif and a C-terminal DNA-binding domain of oligonucleotide/oligosaccharide (OB)-fold; either of which could provide MotB’s ability to bind DNA. DNase I footprinting demonstrates that MotB dramatically alters the interaction of H-NS with DNA in vitro. RNA-seq analyses indicate that expression of plasmid-borne *motB* up-regulates 75 host genes; no host genes are down-regulated. Approximately 1/3 of the up-regulated genes have previously been shown to be part of the H-NS regulon. Our results indicate that MotB provides a conserved function for *Tevenvirinae* and suggest a model in which MotB functions to alter the host transcriptome, possibly by changing the association of H-NS with the host DNA, which then leads to conditions that are more favorable for infection.

## 1. Introduction

Bacteriophage genes of unknown function comprise an abundance of ”dark matter” in the biological universe [1]. One such gene is the phage T4 *motB*, which encodes an early gene product. Although nonessential under normal laboratory conditions, our previous work has demonstrated that a *motB* knockdown yields a 2-fold lower burst than wild type (WT) T4, indicating that it contributes to phage fitness and that purified MotB binds tightly to both unmodified and T4 modified (5-glucosylated, hydroxymethylated cytosine) DNA [2]. Given that *motB* encodes an early protein, these results would be consistent with a role for *motB* in T4 gene expression, as is seen for the T4 early genes *asiA* and *motA*, which are required for middle gene activation [3]. In fact, *motB* was named as a modifier of transcription, from very early work suggesting that a large T4 genomic deletion that removed *motB* affects some T4 middle gene expression [4]. However, our global transcriptomic analyses have demonstrated that a T4 *motB* knockdown does not significantly affect T4 RNA levels [2].

Unexpectedly, our previous work also showed that when bound to DNA, MotB co-purifies with the highly abundant *Escherichia coli* histone-like protein H-NS and its less abundant paralog StpA [2], suggesting that MotB might work by affecting these host proteins. In the bacterial nucleoid, H-NS and StpA, along with other members of the Nucleoid Associated Proteins (NAPs) family, form higher-order nucleoprotein complexes with the host genomic DNA, organizing the DNA within the nucleoid and leading to transcriptional effects through regulation of specific genes [5,6,7,8,9,10].

H-NS binds preferentially to AT-rich sequences [11,12]. Consequently, as xenogeneic genes can display a higher AT content than that of the *E. coli* genome, which is 49.2% AT, H-NS can also protect bacteria from the expression of foreign genes by preferentially binding to their DNA [13,14]. For example, the lytic bacteriophages T7 (52% AT) and T4 (65% AT) would potentially be vulnerable to H-NS repression, and both phages encode anti-H-NS strategies to combat this silencing. The T7 5.5 protein inhibits H-NS by interacting tightly with the central dimerization domain of H-NS, abrogating H-NS oligomerization and the formation of higher-order nucleoprotein complexes [15,16]. In T4, the Arn protein, a structural DNA mimic, sequesters H-NS by binding to its DNA binding domain [17]. However, since the T4 *motB* knockdown does not affect the T4 transcriptome [2], it seems likely that MotB might function in another H-NS-related capacity, rather than to provide a mechanism to relieve H-NS repression of phage gene expression.

To obtain a better understanding of the role of *motB*, we have used *in silico* analyses to investigate the conservation of *motB* and its relationship with domains of known function. We have also used RNA-seq to determine how the presence of MotB affects host gene expression and DNase I footprinting to investigate how MotB affects the interaction of H-NS with DNA. Our results indicate that *motB* is a highly conserved gene among *Tevenvirinae*, suggesting that its DNA binding activity provides an important function among these phages. We find that MotB affects the interaction of H-NS with DNA *in vitro* and that heterologous expression of *motB* in *E. coli* results in the up-regulation of 75 host genes, approximately one-third of which constitute a specific subset of genes that are normally repressed by H-NS. We speculate that DNA-binding by MotB functions to improve phage infection in a heretofore unrecognized way by altering the expression of specific host genes, some of which are normally repressed by H-NS.

## 2. Materials and Methods

### 2.1. Strains

The *E. coli* strain BL21(DE3) [18,19] has been described. LB medium (Quality Biological, Gaithersburg, MD, USA) was used, and cells were grown at 37 °C with shaking at 250 rpm.

### 2.2. DNA

The following plasmids have been described: pNW129, a pACYC-based vector plasmid [20] and pNW129-*motB* (referred to as p*motB*) in which *motB* is located downstream of the inducible promoter P_BAD_ [2]. T4 genomic DNA (gDNA) was purified as previously described [2]. *E. coli* gDNA was purified from an overnight culture of BL21(DE3). *E. coli* cells, harvested by centrifugation at 16,000× *g* for 1 min, were resuspended in lysis buffer containing 0.934 X TE [10 mM Tris-Cl (pH 7.9), 1 mM ethylenediaminetetraacetic acid (EDTA)] buffer and 0.6% sodium dodecyl sulfate (SDS) and then incubated at 37 °C for 1 h. Lysed cells were centrifuged at 16,000× *g* for 5 min to remove cell debris, and the DNA was extracted as described previously for T4 gDNA [2]. After DNA extraction, DNA was purified by ethanol precipitation, resuspended in TE buffer, and incubated at 55 °C for 1 h to fully dissolve the DNA.

To obtain ^32^P-5’-end-labeled fragments containing the *E. coli* promoter for *proV* (P*_proV_*) or the T4 late promoter for gp8 (P_l8_), PCR was performed using purified *E.coli* gDNA or T4 gDNA, respectively, Pfu Turbo polymerase (Stratagene, San Diego, CA, USA), a nonradioactive bottom strand primer, and a 5′-^32^P-labeled top strand primer, which had been previously treated with T4 polynucleotide kinase (Affymetrix, Santa Clara, CA, USA) in the presence of [γ-^32^P]ATP. Primers [purchased from Integrated DNA Technologies (Coralville, IA, USA); Appendix A] annealed such that the resulting fragments were composed of positions −143 to +75 relative to the transcription start site (TSS) of P_l8_ and positions −71 to +201 relative to the TSS P*_proV_*. The PCR-generated DNA was isolated after gel electrophoresis using Elutrap (GE Healthcare, Chicago, IL, USA) and ethanol precipitated.

### 2.3. Proteins

MotB containing a C-terminal His_6_-tag (MotB-His) and H-NS were purified as previously described [2]. After purification, MotB-His and H-NS were stored in MotB storage Buffer [50 mM Tris-HCl (pH 8.0), 0.1 mM EDTA, 50 mM NaCl, 0.01% (*v*/*v*) Triton X-100, 50% (*v*/*v*) glycerol, 0.1 mM dithiothreitol (DTT)] and H-NS storage Buffer [10 mM potassium phosphate (pH 7.5), 200 mM NaCl, 0.1 mM EDTA, 50% (*v*/*v*) glycerol], respectively, at −20 °C.

### 2.4. In Silico Analyses of motB

To search for homologs of MotB, we first used Position-Specific Iterative Basic Local Alignment Search Tool (PSI-BLAST) [21] against the RefSeq database to search for similar sequences in other species. The ”expect threshold” was set to 10^−3^, the PSI-BLAST threshold was set to 0.005, and the BLOSUM62 scoring matrix was set to iterate three times. Only annotated phage proteins (Accession prefix with ”NP_“ or ”YP_”) were then included in a multiple sequence alignment generated by Constraint-based Multiple Alignment Tool (COBALT) [22] for tree building. To calculate distances within the tree, BLOSUM62 was used to measure the similarity between sequences in the alignment, and an Unweighted Pair-Group Method using Arithmetic mean (UPGMA [23]) tree was constructed using JalView 2.11.1.3 [24]. The order of the proteins in the multiple sequence alignment was then re-ordered according to the tree. The taxonomy of the organism’s homologs was obtained from the NCBI Taxonomy Browser: (https://www.ncbi.nlm.nih.gov/Taxonomy/Browser/wwwtax.cgi?id=1198136).

For the MotB homologs that had been assigned to bacterial sources, we searched in the NCBI identical protein database to compile a list of all identical proteins. The geographic location, isolation source, and environmental context for each protein was subsequently determined by identifying the NCBI BioSample associated with the corresponding NCBI Assembly.

To search for possible functional domains within MotB, we performed profile-profile search for remote homologs using the Max Planck Institute Bioinformatic Toolkit HHsearch/HHpred [25,26,27] version 3.2.0 with default settings. The protein sequence and structural family organization of the HHpred top hits were then further investigated using EMBL-EBI Pfam [28] (https://pfam.xfam.org/) version 33.1, SCOP [29] (Structural Classification of Proteins http://scop.mrc-lmb.cam.ac.uk/legacy/ version 1.75) and SCOPe [30,31] (Structural Classification of Proteins-extended https://scop.berkeley.edu/ version 2.07 databases).

### 2.5. DNase I Footprinting

Footprinting was performed as previously described [2]. Briefly, solutions were assembled in a total volume of 10 µL containing 0.05 pmol ^32^P-5’-end-labeled DNA, buffer [40 mM Tris-acetate (pH 7.9), 150 mM potassium glutamate, 4 mM magnesium acetate, 0.1 mM EDTA, 0.1 mM DTT, 0.1 mg/mL BSA], and the indicated amount of MotB-His (4 µL) and/or H-NS (2 µL). For the sequential addition experiments, the protein at the constant concentration was added for 10 min at 37 °C before adding the protein being titrated.

### 2.6. Purification of Total RNA and RNA-seq Analyses

*E. coli* BL21(DE3) cells containing either the vector pNW129 or p*motB* were streaked on 1.5% (*w/v*) LB plates containing 40 µg/mL kanamycin and 0.5% (*w/v*) glucose. Overnight cultures from single colonies were grown in LB containing 40 µg/mL kanamycin and 0.025% (*w/v*) glucose. The next morning inoculums were diluted to an OD_600_ of 0.1 with LB and grown to an OD_600_ of ~0.3, when arabinose [final concentration 0.2% (*w/v*)] was added. At 20 min post-induction (OD_600_ ~0.5), RNA was isolated using method II as described [32]. Total RNA was analyzed on an Agilent 2100 Bioanalyzer (Santa Clara, CA, USA) using the Agilent RNA 6000 Nano Kit to evaluate the quality of the sample.

The cDNA library was prepared using a modified RNATagSeq workflow as previously described [33]. Optimum fragmentation of the total RNA samples in this library was determined to be 3 min at 94 °C in FastAP buffer (Thermo Fisher Scientific, Waltham, MA, USA). The cDNA library was run on a Bioanalyzer using the Agilent High Sensitivity DNA Kit to evaluate the quality of the library. The concentration of the cDNA library was determined by qPCR using the KAPA Library Quantification Kit (Kapa Biosystems, Wilmington, MA, USA) and CFX96 Real-Time PCR Detection System (Bio-Rad, Hercules, CA, USA). Sequencing was performed by the NIDDK Genomics Core facility using a MiSeq system with the single-end 50 bp Sequencing Kit (Illumina, San Diego, CA, USA).

RNA-seq data were processed as previously described using *E. coli* BL21(DE3) (NC_012971.2) as the reference genome [34]. Differential expression between conditions was represented as a fold change, and genes with a fold difference (FD) ≥ 2, an adjusted p-value ≤ 0.05, and mean reads ≥ 5 were considered significant. RNA-seq data is available in the National Center for Biotechnology Information (NCBI) database (GEO #GSE152170) and in Appendix A. Visualization of the transcriptomics data into representative categories was performed using a modified version of the EcoCyc Omics Dashboard tool (ecocyc.org) as described [35].

## 3. Results

### 3.1. In Silico Analyses Predict That MotB Contains Both a KOW and an OB-Fold Domain

To elucidate possible functions for MotB, BLAST and PSI-BLAST were first performed in the hope of finding proteins of known function with high sequence similarity. However, this analysis did not reveal any such characterized proteins; only *motB* homologs within the *Tevenvirinae* subfamily of *Myoviridae* were observed (detailed below). Consequently, we employed a predictive structural homolog search using HHpred, and we investigated domain organizations further using the Pfam, SCOP, and SCOPe databases. The top 10 HHpred hits for MotB are listed in Appendix A. These analyses predicted that MotB contains two domains (Figure 1), an N-terminal domain (NTD) related to the Kyprides-Onzonis-Woese (KOW) motif [36] and a C-terminal domain (CTD) related to the DNA-binding domain of oligonucleotide/oligosaccharide (OB)-fold [37].

Within the Pfam classification, 8 of the top 10 hits, which aligned to MotB^NTD^ residues 6 to 45, contained a predicted KOW-like domain. Furthermore, all of the HHpred hits with a probability >50% indicated that the MotB^NTD^ belongs to the all-beta, b.35.5 SCOPe Superfamily named “Translation proteins SH3-like domain”, and many of these family members also contain a KOW motif. KOW domain proteins, which are found in all domains of life, are known to mediate both protein-protein and protein-nucleic acids interactions [36] and include the highly conserved transcriptional elongation factors human hSpt5 [38] and bacterial NusG [39,40] as well as various ribosomal proteins.

The MotB^CTD^ is predicted to belong to the SCOPe Family b.40.4.5, cold shock DNA-binding domain of OB-fold. Members of this family have also been shown to mediate protein-DNA, protein-RNA, and protein-protein interactions [42]. In this case, the closest hits to MotB^CTD^ were the translation initiation factor 5A found in two archaeal species, 1BKB from *Pyrobaculum aerophilum* and 2EIF from *Methanococcus jannaschii*.

We conclude that MotB is likely a two-domain, N-terminal KOW/C-terminal OB-fold protein. As both KOW and OB-fold motifs are known to be involved in protein-nucleic acids interactions, either of the domains could account for MotB DNA binding activity.

### 3.2. MotB Is Highly Conserved among T-Even Phage

As the *motB* gene lies in a large, nonessential region of the T4 genome [4], it was important to determine its level of conservation. Consequently, we performed a homology search, as described in Materials and Methods. We found 112 homologous proteins from 54 species (Appendix A). All but 10 of these proteins are within genomes found in the Myoviridae family/Tevenvirinae subfamily of the genera *Tequatrovirus, Gaprivervirus, Dhakavirus, Mosigvirus*, or unclassified. The other 10 hits were found within bacterial genomes having a WP_ accession prefix, indicating a non-redundant match across multiple strains and species (Appendix A). These hits, which arose from proteins found in specific strains of *E. coli*, *Salmonella enterica*, and *Bacillus cereus,* were further investigated by compiling a list of all identical proteins and determining geographic location, isolation source, and environmental context of each protein indicated in their corresponding NCBI BioSample entry (Appendix A). The identical protein list revealed that 2 of the hits contained proteins from both bacterial and phage origin, whereas the remaining 8 hits presumably only contained proteins from bacteria. However, the BioSample entry for these proteins indicated a heterogeneous sample source, including cell culture from a poultry farm, ground meat, humans, cattle, and phage infections. Due to the heterogeneity of these assemblies and our suspicion that they arose from phage contamination, we removed the 10 ‘bacterial’ hits from the multiple sequence alignment before continuing.

We generated a multiple sequence alignment of the remaining 102 phage homologs (Appendix A). This analysis showed that the proteins fall into four distinct groups: Category 1 (names in magenta), 51 proteins that are highly similar to MotB in both the NTD (KOW) and the CTD (the cold shock DNA-binding domain of OB-fold region); Category 2 (names in blue), 25 proteins that retain similarity in the CTD, but have a different NTD region; Category 4 (names in green), 25 proteins that have similarity in the CTD, but have another distinct NTD motif; and Category 3 (name in grey), 1 hypothetical peptide of only 29 amino acids in phage RB69 that aligns nearly perfectly to the highly conserved CTD core motif found in categories 2 and 4. Additional entries for Category 3 may have been missed since a protein of such a small size might not have been automatically annotated.

Category 2 contains another T4 early protein of unknown function, MotB.1, which is encoded by the gene immediately upstream of *motB*. The T4 MotB and MotB.1 proteins share a conserved CTD but have different NTD regions (Appendix A). The high similarity between the CTD’s of MotB and MotB.1 suggests that within *Tevenvirinae* phages, one gene might have arisen from a duplication of the other. Categories 2, 3, and 4 represent proteins that have more homology within their CTDs to MotB.1 than to MotB. Besides T4, 40 other phage genomes had both a *motB* and *motB.1*, including one phage, RB69, which had two copies of *motB*, 1 copy of *motB.1*, and the 29 aa peptide in Category 3 (Appendix A).

The sequence alignments were then used to calculate distance and generate a UPGMA tree. The tree shows that the MotB group (Category 1, colored in magenta) represents a distinct branch from the branches containing the MotB.1 group (Categories 2-4 colored in blue, grey, and green, respectively) (Figure 2). There was no clear correlation between the particular tree branch and the specific phage genera (Appendix A). In a previous classification of *Myoviridae* subfamilies, MotB along with the T4 Pin and T4 ModA proteins was used to subdivide the *Teequatrovirinae* (i.e., *Tevenvirinae*) subfamily into ”T4-like” and the ”KVP40-like” groups [43]. Using this classification, all of the homologs were within the ”T4-like” phage group.

Our phylogenetic analyses demonstrate that despite its position within a region of the T4 genome that is nonessential under normal laboratory conditions, *motB* and the related gene *motB.1* are highly conserved within *Tevenvirinae*. It seems highly likely that *motB*, and the DNA binding activity encoded by its gene product, fulfills an important biological role for the phage.

### 3.3. MotB Affects the Interaction of H-NS with DNA

Despite its ability to bind both unmodified and modified DNA, our previous work indicated that MotB is not involved in the regulation of the T4 transcriptome; instead, we made the unexpected finding that during purification, MotB co-purifies with DNA and the NAP proteins H-NS and its analog StpA [2]. H-NS primarily interacts nonspecifically with AT-rich regions [11,12]. Such regions are frequently found within genes that have been horizontally transferred into the *E. coli* genome. Consequently, the H-NS/DNA interaction typically silences the expression of these xenogeneic genes [12]. However, H-NS is also known to have some preferred host sites, including those downstream of the TSS of the *E. coli proV* gene [44]. Thus, to investigate whether MotB can affect the interaction of H-NS with DNA, we performed DNase I footprinting (Figure 3A) with a sequence surrounding the *proV* promoter P*_proV_*(Figure 3B).

As expected, increasing concentrations of H-NS alone resulted in protected regions downstream of the P*_proV_* TSS that were similar to previously obtained footprints of H-NS at the *proV* gene [[44,45]; Figure 3A, lanes 3–6, protected regions indicated by red bars]. In contrast, increasing concentrations of MotB alone resulted in no significant change in the footprint, until a ratio of ~2.4 MotB monomers: 1 bp was achieved, at which point protection of nearly the entire region was observed (Figure 3A, lane 10). This result was similar to our previous MotB footprinting results using a different fragment of DNA [2].

We then investigated the effect of adding MotB and H-NS together. We kept the amount of either H-NS or MotB at 16 pmol (a protein:bp ratio of 1.2:1; Figure 3A, lanes 5 and 9, respectively) and varied the amount of the other protein from 4, 8, 16, to 32 pmol. In one case, we added the proteins together (Figure 3A, lanes 11–18), while in the other case, we added the proteins sequentially, incubating the first protein with the DNA for 10 min at 37^o^ C before adding the second protein (Figure 3A, lanes 20–27.) Two results were striking. First, despite which protein was added first, the footprints were more like those seen with the highest level of MotB alone (32 pmol, lane 10) rather than those seen with high levels of H-NS alone (lanes 5 and 6). Second, the level of protein needed to achieve complete protection was now observed with 16 pmol of MotB rather than 32 pmol (compare lane 9 having 16 pmol of MotB alone with lanes 13, 15–18, 22, 24–27). The most complete protection level was observed in the presence of both H-NS and MotB, suggesting that binding could be cooperative. Together these results suggested that the protein/DNA interaction mediated by MotB dominates over that of H-NS and that perhaps H-NS aids MotB in its interaction with DNA. We conclude that the binding of H-NS to *proV* is dramatically altered by the presence of MotB.

In a similar manner, we also investigated the binding of H-NS to a fragment containing the promoter for the T4 late gene *8* (positions −143 to +75 relative to the TSS). Our previous work had indicated that at the very start of late gene expression there is a minor decrease in gene 8 RNA levels in a *motB* knock-down infection, but this was not significant later in infection when late gene expression is high [2]. The DNase I footprints observed with this DNA (Appendix A) yielded similar results to those observed with P*_proV_*. Increasing levels of H-NS resulted in the protection of portions of the DNA, while MotB gave either no protection at lower levels or total protection at a ratio of 2.9 MotB:1 bp DNA. Again, in the presence of both proteins, the patterns resembled that of MotB, not H-NS. These results again suggested that MotB alters H-NS binding.

### 3.4. Expression of motB Results in Up-Regulation of 75 Host Genes, a Subset of Which Are Repressed by the Histone-Like Protein, H-NS

Given that MotB altered the interaction of H-NS with DNA *in vitro,* we wondered whether the heterologous expression of *motB* would affect host gene expression *in vivo*. To investigate, we performed RNA-seq analyses of RNA isolated after expression of *motB* in exponentially growing *E. coli*. As T4 is classically cultured in *E. coli* B, we used BL21(DE3) as the host. We chose 20 min after *motB* expression for the RNA isolation in order to observe early changes, and thus more direct effects of MotB. Importantly, at this time point, the level of protein is not detectable by SDS-PAGE, and the cells are still growing similarly to those containing the vector plasmid [2].

Our RNA-seq analyses indicated that 75 host genes were significantly up-regulated with a FD ≥ 2, adjusted *P*-value ≤ 0.05, and mean reads ≥ 5 (Appendix A). No genes were significantly down-regulated. More than 50% of the affected genes were associated with either the defective λ lysogen DE3 or other prophages/cryptic prophages present in the chromosome with many of the genes located downstream of the λ DE3 P_L_ or P_R_ promoters. Several of the phage genes were up-regulated more than a 1000-fold: λ*cI*, λ*cII*, λ*O*, λ*S*, λ*R*, DLP12 *ybcV,* and DLP12 *ybcW*. The significant MotB-mediated up-regulation of genes from the λ DE3 lysogen genes and other prophages within the *E. coli* BL21(DE3) chromosome could be a likely explanation for the cell death that accompanies *motB* expression in this strain [2].

To visualize our transcriptomic data, we used a modified version of the “Pathway Tools Omics Dashboard” (ecocyc.org) (Appendix A), in which a series of panels present genes broadly related to cellular systems. Genes that responded to MotB were found in various pathways (biosynthesis, central dogma, cell exterior and response to stimulus), and the gene with the greatest FD (apart from the DE3 or prophage genes) was *zapA*. Overexpression of *zapA* has been shown to increase cell length due to the incorrect localization of FtsZ and ZapB, proteins that are integral to Z-ring formation and cell division; however, overproduction of ZapA does not alter growth rate [46]. In addition, there was up-regulation of three genes that are associated with acid resistance (*evgS*, encoding the acid-sensing histidine kinase EvgS; *slp*, encoding a starvation lipoprotein; and *gadE*, encoding the acid resistance transcriptional activator GadE) [47,48] (Appendix A). However, the percentages of activated genes in any particular category were not significant, suggesting that MotB does not target a particular host pathway. In addition, although previous work has suggested that H-NS can affect CRISPR-Cas regulation [49], expression of the two known CRISPR loci present in BL21(DE3) [50] was not affected by the presence of MotB.

Despite the fact that no particular gene(s) whose overexpression is known to improve T4 infection was identified, we did find a correlation between affected host genes and the H-NS regulon. Previous global transcriptomic analyses have identified 172 (~4%) and 583 (~13%) *E. coli* genes that are dysregulated in an *hns* deletion or a double *hns*/*stpA* deletion, respectively [51]. Other work has identified additional genes that are affected in the presence of the T7 5.5 protein, which interferes with H-NS oligomerization, or by the overexpression of a C-terminally truncated *hns*, which lacks the DNA binding domain and interferes with WT H-NS function [16,52]. We found that 30% of the genes affected by MotB (22 genes) were assigned as H-NS-regulated genes in at least one of these analyses (Appendix A). This included *proV*, whose DNase I footprint pattern by H-NS was altered by MotB (Figure 3). Furthermore, 5 additional genes of the biotin operon are known to be indirectly affected by H-NS through the up-regulation of the λ*Q* gene [53]. Taken together, these results are consistent with the idea that MotB binding to host DNA interferes with specific H-NS repression of the host genome.

## 4. Discussion

The ongoing arms race between bacteria and lytic phages involves early phage genes that are needed to take over the host and host genes that can respond to this challenge. In T4, early gene expression, which commences within 1 min post-infection, produces both these ‘takeover’ proteins as well as factors needed for T4 gene expression. The majority of the early T4 genes are nonessential, suggesting that they optimize the infection and/or are only required under certain conditions. However, the functions of most of these products have not been characterized.

The T4 MotB protein is one such early product. We have previously shown that *motB* improves phage fitness and is toxic when expressed heterologously in either *E. coli* B or K12 strains [2]. Although MotB binds tightly to DNA and very early work suggested that *motB* might be involved in regulating phage transcription [4], the levels of early, middle, or late RNA during a T4 infection are not significantly affected by a *motB* knockdown [2]. This suggests that the DNA binding activity of MotB serves a purpose other than affecting the T4 transcriptome. Our *in silico* analyses indicate that MotB is a highly conserved protein within *Tevenvirinae* and predict that MotB contains an N-terminal KOW domain and a C-terminal OB-fold. Either of these motifs could confer the strong DNA binding activity associated with the protein. Given these findings, we conclude that the MotB DNA binding activity provides an important conserved function for these phages, despite its location within a ”nonessential” region of the T4 genome.

Curiously, the OB-fold motif within the CTD of MotB is related to the CTD within another early protein of unknown function, MotB.1. In the case of T4, MotB.1 is encoded by a gene that lies immediately upstream of *motB*. Several phage genomes carry more than one copy of either *motB* or *motB.1*. The function of *motB.1* is not known, but like *motB*, it is an early gene located in a nonessential region of the T4 genome.

In a search for the biological function of MotB, we have investigated how its DNA binding activity might aid the phage. We found that in the presence of DNA, MotB copurifies with the *E. coli* NAPs: H-NS and its paralog StpA [2]. Bacterial NAPs are responsible for the organization of genomic DNA through the formation of high-order complexes that condense DNA, impacting major processes including replication, recombination, repair, and transcription [5,6,7,8,9,10,54]. As xenogeneic genes are often AT-rich, H-NS can also serve to repress foreign genes acquired by horizontal transfer and has been implicated as a host defense mechanism against both lytic and lysogenic phages [55].

Several phage- and plasmid-encoded genes are known to interfere with this H-NS silencing of their own genomes (reviewed in [13,56]). Some of these, such as the T7 5.5 protein, which disrupts H-NS oligomerization [15,16] and the T4 Arn protein, which binds to the DNA-binding domain of H-NS [17], interact directly with H-NS. Other proteins, such as the plasmid-borne VirB of *Shigella flexneri* [56], ToxT encoded within the pathogenicity island of *Vibrio cholerae* [57], and Ler encoded within the pathogenicity island LEE1 of pathogenic *E. coli* [58], work by binding to regions of the xenogeneic DNA, removing H-NS and relieving silencing. In our case, we have only observed an association of MotB with H-NS (and StpA) in the presence of DNA. Our repeated attempts to observe a MotB/H-NS protein-protein interaction that does not require DNA have been unsuccessful. Thus, MotB would appear to be more like the latter group of proteins. However, we have not identified any specific sites or regions of MotB binding.

Nearly 1/3 of the genes upregulated in the presence of MotB have previously been assigned as part of the H-NS regulon, including *proV*, whose repression by H-NS is well-characterized [44,45], as well as multiple prophage genes and genes within the DE3 lambda lysogen. In fact, the most dramatic upregulation involves the phage genes. Although these genes are not normally considered part of the H-NS regulon, previous transcriptomic work using BL21(DE3) has demonstrated that the presence of specific H-NS antagonists (T7 5.5 protein or a C-terminally truncated H-NS) dysregulates their expression (indicated in Appendix A; [16]). Furthermore, earlier studies have indicated that a T7 5.5 mutant does not plate on a λ lysogen [59,60], leading to the speculation that a T7 5.5 interaction with H-NS affects the expression of the λ genes [16]. Consequently, we conclude that a substantial effect of MotB involves the upregulation of BL21(DE3) genes that are normally repressed by H-NS. However, this effect is specific to certain loci. For example, the *bgl* operon, known to be repressed by H-NS [55,61], is not significantly affected. In addition, although work has shown that H-NS represses CRISPR-Cas regulation in *E. coli* ([62] and references therein), we observed no significant change in the expression of the two known CRISPR loci present in BL21(DE3) [50] in the presence of MotB.

We speculate that DNA-binding by MotB functions to improve phage infection in a heretofore unrecognized way, by altering the expression of specific host genes, some of which are normally repressed by H-NS. We base this idea on the following observations. First, previous reports indicate that MotB is a highly abundant protein, whose synthesis begins 1 min after infection and continues for several min onward [63,64]. Second, we have shown that MotB binds tightly and with similar affinity to both unmodified host DNA and T4-modified DNA [2]. Consequently, until T4 replication begins (5-6 min post-infection), we expect that a large amount of MotB present within the cell will associate with host DNA. Although T4 nucleases will begin to degrade the host DNA shortly after infection [65], a considerable level of host DNA will remain for several min [66], and we expect that this DNA will be associated with the abundant host NAPs, including H-NS. Our results suggest that the binding of MotB to the host genomic DNA will alter this association, which leads to the dysregulation of specific host genes and results in a better host for infection. The specific host genes, whose dysregulation is helpful for T4, have not yet been identified. However, it should be noted that our analyses, which were performed under standard laboratory conditions, would not have identified host genes that would aid in more biologically relevant growth conditions. In addition, one or more of the several up-regulated genes, which have not been identified as part of the *hns* regulon, may provide an unknown benefit for T4. Similarly, although we have previously found no significant change in T4 transcription in a *motB* knockdown infection of BL21(DE3) under standard laboratory conditions [2], we cannot rule out the possibility that MotB may relieve H-NS repression of the AT-rich T4 DNA and thus, regulate transcription of T4-encoded genes under specific growth conditions and/or in different strains.

Our *in vitro* footprinting data suggests that MotB binds cooperatively with H-NS at the region surrounding the start of *proV* (Figure 3). Thus, one might expect that *proV* should remain repressed, rather than up-regulated, as we observe *in vivo*. However, the footprints also show that when both proteins are present, the overall binding pattern of H-NS is altered to resemble that of MotB alone. We speculate that the presence of MotB mediates changes in the binding pattern and/or the mode of H-NS binding, which then leads to the dysregulation of H-NS mediated repression. This could arise from global changes in nucleoid structuring since H-NS interference is thought to occur through changes in local nucleoid structure as additional non-specific DNA binding proteins (e.g., Fis, IHF, HU) augment H-NS-mediated silencing through changes to protein-DNA complex (reviewed in [14]). Furthermore, supercoiling and osmolarity have been shown to alter H-NS activity and upregulate the *proU* locus [67,68]. It has been proposed that changes in supercoiling and DNA topology may only impact certain promoters [14], which could explain the upregulation of only a subset of the H-NS regulon by MotB.

The finding that the synthesis of MotB continues well into the period of late gene expression/ DNA replication is atypical since the synthesis of many T4 early gene products ceases quickly [69]. Thus, it seems likely that MotB may also serve a function later in infection, perhaps for phage replication and/or packaging. Ongoing work centers on determining the effect of MotB on these phage processes as well as on host gene expression throughout infection.

## Figures and Tables

**Figure 1 viruses-13-00084-f001:**
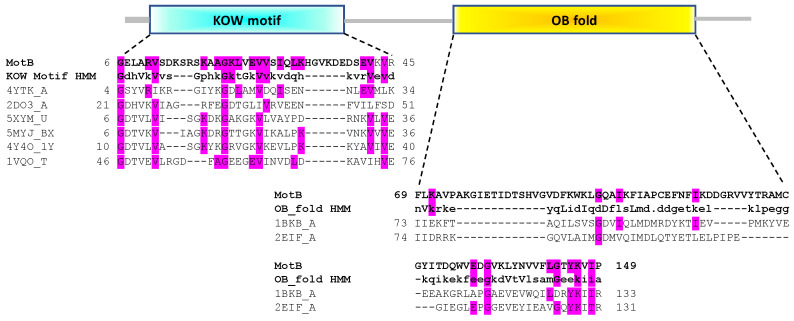
Domain organization of MotB predicts an N-terminal KOW and a C-terminal OB-fold domain. The Pfam Hidden Markov Model (HMM) consensus sequences of the KOW and OB-fold motifs are aligned with MotB residues 6 to 45 and 69 to 149, respectively, along with some other top hits for MotB found by HHsearch/HHpred (Appendix A). For the KOW domain, these hits are PDB ID 4YTK, the KOW1-Linker1 domain of *Saccharomyces cerevisiae* Spt5 transcription elongation factor; 2DO3, the third KOW motif of human Spt5; and 5XYM, 4Y4O, 5MYJ, and 1VQO, ribosomal proteins of the bacteria *Mycobacterium smegmatis, Thermus thermophilus* and *Lactococcus lactis* and the archaean *Haloarcula marismortui,* respectively. For the OB-fold domain, these are 1BKB and 2EIF, the translation initiation factors 5A of the archaea *Pyrobaculum aerophilum* and *Methanocaldococcus jannaschii*, respectively. In each case identical residues within MotB and the consensus and/or MotB and the hits are highlighted in magenta. The alignment for the C-terminal portion of the third KOW motif of human Spt5 has been previously reported [41]. More information about the top 10 hits is given in Appendix A.

**Figure 2 viruses-13-00084-f002:**
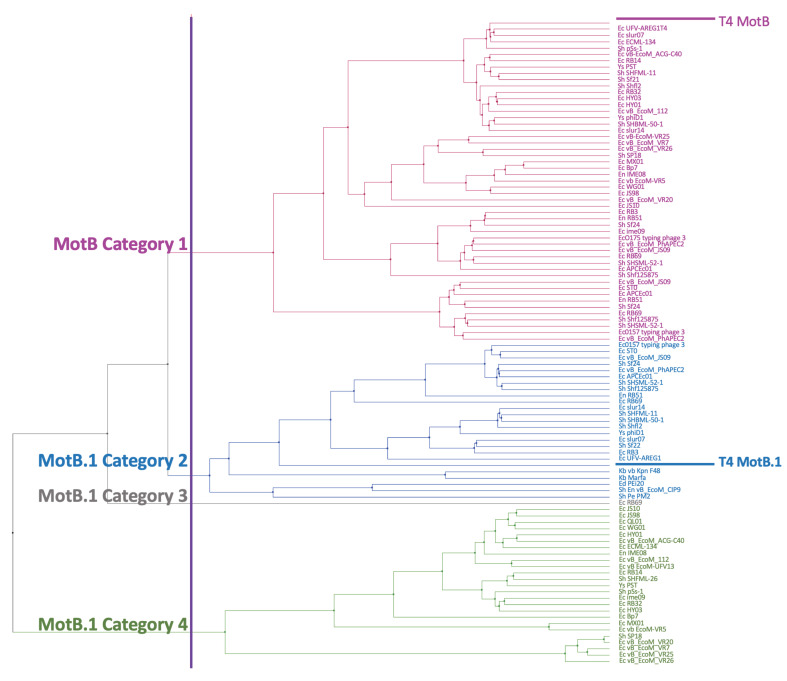
MotB phylogenetic tree. A multiple sequence alignment generated by COBALT for the 102 PSI-BLAST phage hits was used to construct the UPGMA tree. The purple vertical line indicates where cutting of the tree generates 4 branches, separating MotB homologs (Category 1, magenta) from the related MotB.1 homologs (Category 2, blue; Category 3, grey; Category 4, green). Names on the right indicate the host (Ec: *Escherichia*, Sh: *Shigella*, En: *Enterobacter*, Ys: *Yersinia*, Ed: *Edwardsiella*, Kb: *Klebsiella*) followed by the phage name. See Appendix A for details about each protein; order here is same as order in Appendix A.

**Figure 3 viruses-13-00084-f003:**
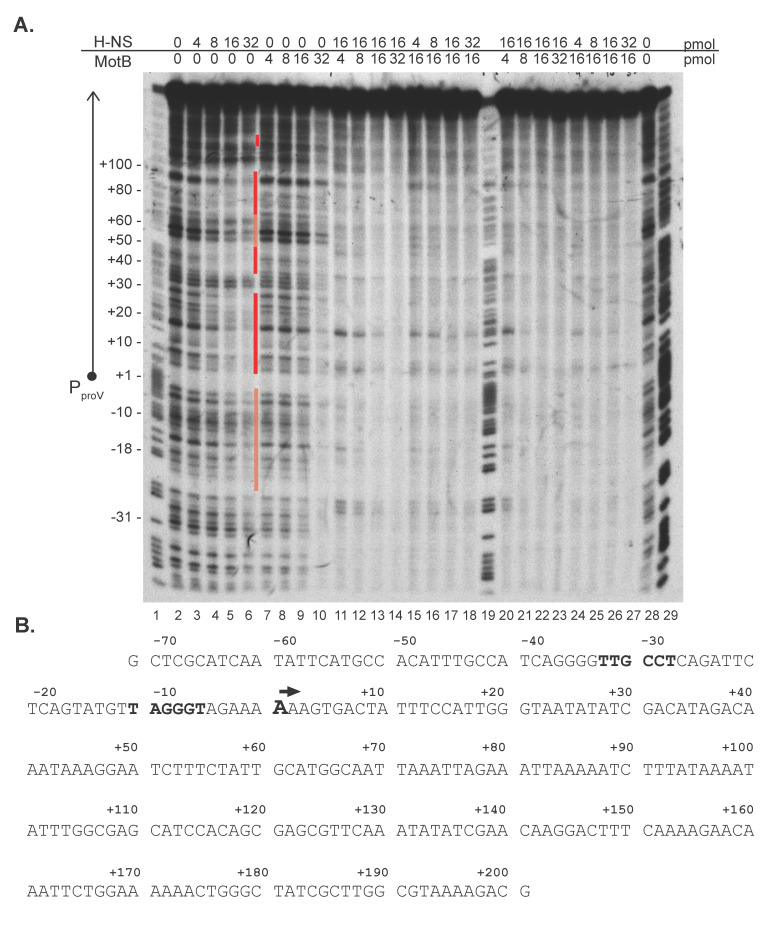
Representative DNase I footprint of MotB and/or H-NS at P*_proV_* and the P*_proV_* sequence. (**A**) DNA surrounding the P*_proV_* promoter (positions −71 to +201; 0.05 pmol DNA; 13.6 pmol total bp; 5’-^32^P labeled on the nontemplate strand) was incubated with the indicated amount of MotB-His and/or H-NS and treated with DNase I before electrophoresis on a 5% (w/v) polyacrylamide, 7 M urea denaturing gel. A schematic of the P*_proV_* promoter region is shown to the left of the gel, and the positions corresponding to the DNA sequence determined from the G + A ladder are indicated. Regions protected or partially protected by H-NS are indicated by the dark and light red bars, respectively, to the right of lane 6. Lanes 1, 19, 29: G + A ladder; lane 2, 28: no protein control; lanes 3–6: 4 pmol, 8 pmol, 16 pmol, and 32 pmol H-NS, respectively; lanes 7–10: 4 pmol, 8 pmol, 16 pmol, and 32 pmol MotB, respectively; lanes 11–14 and 20–23: 16 pmol H-NS with 4 pmol, 8 pmol, 16 pmol, and 32 pmol MotB, respectively; lanes 15–18 and lanes 24–27: 16 pmol of MotB with 4 pmol, 8 pmol, 16 pmol, and 32 pmol H-NS, respectively. For lanes 11–18, H-NS and MotB were added together; for lanes 20–27, the protein whose concentration did not change was added first and incubated with the DNA for 10 min at 37^o^ C before the addition of the second protein. (**B**) Nontemplate sequence of the P*_proV_* promoter (−71 to +201). The +1 TSS is indicated by the black arrow and the +1 TSS and the −35 and −10 elements are in bold.

## Data Availability

RNA-seq data is available in the National Center for Biotechnology Information (NCBI) database (GEO #GSE152170) and in Appendix A.

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
