# Peer review of "Overexpression of the Bacteriophage T4 motB Gene Alters H-NS Dependent Repression of Specific Host DNA"

_viruses, 2021, doi:10.3390/v13010084_

Round 1

Reviewer 1 Report

In this manuscript (viruses-1060106) authors present data obtained by investigation of the protein of unknown function MotB of phage T4. They show that MotB has two putative DNA-binding domains, interacts with the host and phage DNA nonspecifically, and alters the expression of a subset of host genes when it is expressed from plasmid. In addition, authors show that MotB is highly conserved among related phages indicating that it pays an important function for these phages.

The data is clearly presented, and the manuscript is well written, so I have only minor comments and suggestions that might improve it.

  1. The main conclusion of the work presented here is that MotB protein functions to alter H-NS repression of the host genome leading to the better cellular environment for phage infection. However, I think that authors should not rule out the possibility that MotB might also act as a protector of A/T-rich phage DNA from H-NS repression. This possibility can be supported by these facts: a) MotB is an abundant early protein, which can interact with the modified phage DNA as well; b) H-NS prefers A/T-rich sequences for binding and repression of foreign genes; c) authors found that “protein/DNA interaction mediated by MotB dominates over that of H-NS and that perhaps H-NS aids MotB in its interaction with DNA“ (L297-L299 and Figure 3) and “in the presence of both proteins, the patterns (of DNA protection) resembled that of MotB, not H-NS“ (L308, Figure S2). Therefore, I would suggest authors to consider this possibility as well.
  1. I noticed several inaccuracies in the text that should be corrected:
  1. L89 “sulfide“ or “sulfate“?
  2. L305 “Figure S1“ should be “Figure S2“;
  3. I did not find the reference for Bouffartigues et al., 2007, which was cited in the Table S2;
  4. In the column “Species“ of Table S5 “Salmonella“ should be written “Salmonella enterica“.

Based on these comments, I suggest a minor revision of the manuscript.

Reviewer 2 Report

Review

This is an interesting article that advances the understanding of the function of T4 protein MotB by combining several complementary approaches, such as in silico domain organization and phylogenetic analysis, DNAse I footprinting with purified proteins and RNAseq of cells expressing MotB in the absence of T4 infection. The work is solid and well performed.

Major comments

  1. In my opinion the tittle leads to a generalization that is not supported by the data. Since MotB effects have been detected only on a subset of host genes, the tittle should be:

“Overexpression of the bacteriophage T4 motB gene alters H-NS dependent repression of specific host genes”

  1. As the authors point out, the in silico analysis supports some important/conserved function of MotB, since this protein is conserved in the T-even phage family. However, as in other cases, this function may only be evident in the environment and not under lab conditions. And that may be part of the explanation as why it is not easy to connect the effects of the MotB expression with the regulation of host genes that would bring some benefit to the phage infection. I think this possibility should be commented in the discussion.
  1. The footprinting results point clearly to certain cooperativity in the binding of H-NS and MotB to DNA. This is typical of proteins that bind together to the DNA, and, if this binding covers the promoter, it will normally lead to transcription repression. However along the paper the main effect that is proposed is up-regulation, which is more difficult to reconcile with the footprinting results. A possible reconciling explanation is that the lower amount of protein expressed in vivo would lead not to cooperative binding but to H-NS binding interference and that could result in up-regulation. The authors suggest MotB alters the binding of H-NS, but this alteration should be better explained, and compared with some similar case in the literature.
  1. By looking at the RNAseq data, it is acceptable that there is certain correlation between H-NS and MotB regulated genes. However, the magnitude of the regulation is small 5-fold at the most and in many cases around 2-fold, that is near the detection limit. So, under the experimental conditions, these are subtle effects.

On the other hand, there are 17 host genes that do not appear to be regulated by H-NS, so the possibility should be admitted that MotB regulates the host expression by a non-H-NS dependent mechanism, unless they could be attributed to secondary effects.

But, importantly, there are huge effects of MotB on the lambda prophage transcription that go just mentioned. This strong up-regulation of lambda genes is independent of the host SOS response that is the standard trigger for lambda lytic activation, given that we do not observe any host SOS genes up-regulated. Also, since H-NS has not been strongly connected to lambda phage regulation, the simplest explanation is that MotB is interfering more or less directly with the binding, probably, of the lambda cI repressor, and perhaps other lambda regulators. Interestingly, this up-regulation occurs also with other prophages present in the host, and this fact could point to some inter-phage interference by MotB, although the general mechanism is unclear. All this has not been commented almost at all and is one of the most salient results of the paper. It should be adequately commented in discussion again comparing it with similar cases. In future work I would suggest doing footprinting with the PR-PL region of lambda to see what kind of interaction/mechanism can be observed.

If the above points were attended the article should be ready for publication.

Minor points

I think that a more appropriate tittle should be “Overexpression of the bacteriophage T4 motB gene alters H-NS dependent repression of specific host genes”

P2 L48. Specify which MotB mutation

P6 L228. Substitute “may or may not have been annotated in the genome” by “might not have been automatically annotated.”

P6 L228. “within the genome” is redundant, eliminate.

P9 L297. To me, the most complete protection level was only observed in the presence of both H-NS and MotB, and the binding is cooperative. A sentence reflecting those facts should be added.

P10 L348. It should say: “… were assigned as H-NS regulated genes…”
